# Advances in the Biosynthesis and Molecular Evolution of Steroidal Saponins in Plants

**DOI:** 10.3390/ijms24032620

**Published:** 2023-01-30

**Authors:** Yi Li, Huan Yang, Zihao Li, Song Li, Jiaru Li

**Affiliations:** State Key Laboratory of Hybrid Rice, College of Life Sciences, Wuhan University, Wuhan 430072, China

**Keywords:** steroidal saponins, biosynthesis, regulation, distribution, evolution

## Abstract

Steroidal saponins are an important type of plant-specific metabolite that are essential for plants’ responses to biotic and abiotic stresses. Because of their extensive pharmacological activities, steroidal saponins are also important industrial raw materials for the production of steroidal drugs. In recent years, more and more studies have explored the biosynthesis of steroidal saponins in plants, but most of them only focused on the biosynthesis of their molecular skeleton, diosgenin, and their subsequent glycosylation modification mechanism needs to be further studied. In addition, the biosynthetic regulation mechanism of steroidal saponins, their distribution pattern, and their molecular evolution in plants remain unclear. In this review, we summarized and discussed recent studies on the biosynthesis, molecular regulation, and function of steroidal saponins. Finally, we also reviewed the distribution and molecular evolution of steroidal saponins in plants. The elucidation of the biosynthesis, regulation, and molecular evolutionary mechanisms of steroidal saponins is crucial to provide new insights and references for studying their distribution, diversity, and evolutionary history in plants. Furthermore, a deeper understanding of steroidal saponin biosynthesis will contribute to their industrial production and pharmacological applications.

## 1. Introduction

Saponins are a kind of natural secondary metabolite with various structures and functions, and are widely distributed in plants [1]. They are a group of compounds with complex chemical structures composed of a triterpenoid or steroidal aglycone molecular skeleton conjugated to a sugar moiety, classed as triterpenoid saponins and steroidal saponins, respectively [1]. Because of the combination of a hydrophobic aglycone skeleton and hydrophilic sugar chain, these compounds have water surface tension, amphiphilic (i.e., hydrophilic and hydrophobic), foaming, and emulsifying properties, which is why they are named saponins, derived from *sapo*, i.e., soap [1,2,3]. Saponins isolated from plants are considered to have a variety of properties, often acting to protect plants from pathogens and plant-feeding insects. For humans and animals, saponins can affect cell membrane permeability or the lysing of cells, have hypoglycemic activity, and can also affect cholesterol metabolism; they, therefore, have a wide range of applications in food, cosmetics, pharmaceuticals, and other fields [1,4].

Although saponins are ubiquitous in plants, the steroidal saponins discussed in this paper are mainly distributed in monocots, such as the Dioscoreaceae (*Dioscorea zingiberensis*), Melanthiaceae (*Paris polyphylla*), and Asparagaceae (*Asparagus officinalis*) [1,3]. Steroidal saponins isolated from plants can be divided into the following five types according to their molecular skeleton structures:(1)Spirostanol saponins: a hexacyclic ABCDEF-ring system characterized by an axial methyl or hydroxymethyl on the F ring (C-27) [5,6,7,8,9,10] (Figure 1A). The core aglycone of spirostanol saponins has cis- or trans-fusion between ring A and ring B, or has double bonds between ring C-5 and C-6 [7]. The common spirostanol saponins isolated from plants include dioscin, gracillin, and trillin [5,9].(2)Furostanol saponins: a pentacyclic ABCDE ring with a sixth open F ring [5,6,7,8,9,10] (Figure 1B). The common furostanol saponins include parvifloside, protogracillin, and protodioscin [5]. Furostanol saponins usually have 25(R) and 25(S) structures, or are saturated at the C-20 (22) or C-22 (23) positions on the open F ring [5,6]. The two sugar chains of furostanol saponins are usually connected at the C-3 and C-26 positions, while the β-glucoside at the C-26 position will produce a closed ring reaction under the catalysis of glycosidase and convert it into spirostanol saponin [8,11].(3)Cholestane saponins: produced by the oxidative cracking of the C-22/C-23 bond of the aglycone skeleton [7,8,9] (Figure 1C). Cholestane saponins (such as Anguivioside XV and Smilaxchinoside D) have been found in some *Smilax* species [7]; a homo-cholestane saponin with aromatic ring E (such as Paris pseudoside A and B) has also been found in some *Paris* species [8].(4)Pregnane saponins: a tetracyclic ABCD-ring system [5,7] (Figure 1D). They may be biosynthesized via the oxidative cleavage of the double bond between C-20 and C-22 in the structure of furostane [7,9]. Pregnane saponins isolated from plants include Spongipregnoloside A/B/C/D/E, Trinervuloside A, Riparoside B, and Timosaponin J/K [5,7].(5)Isospirostanol saponins: monosaccharide chain saponins. Their unique feature is the equatorial methyl or hydroxymethyl on the F ring (C-27) [7] (Figure 1E). Isospirostanol saponins include the following types: dehydrogenation between C-5 and C-6, carbonylation at C-6, hydroxylation at C-17 or C-27, and cis–trans fusion between the A ring and B ring [7].(6)Polyhydroxylated saponins: a six-ring system [8] (Figure 1F). They have hydroxyl groups at the C-1 position, and may also be hydroxylated at the C-1, C-3, C-21, C-23, and C-24 positions [8,12,13]. Moreover, the fructose at the C-24 position of the aglycone is unique in *Paris* plants [8].(7)Pseudospirostanol saponins, a kind of tetrahydropyran F ring, are rare in plants, and include nuatigenin and isonutigenin isolated from *Paris* species [8] (Figure 1G).(8)Pennogenin saponins are also rare in plants (Figure 1H). If diosgenin is also hydroxylated at C-17, C-23, C-24, and C-27, it can form pennogenin or hydroxy-penogenin saponin, such as polyphyllin D, Paris VI, and Paris VII isolated from *Paris* species [8].

**Figure 1 ijms-24-02620-f001:**
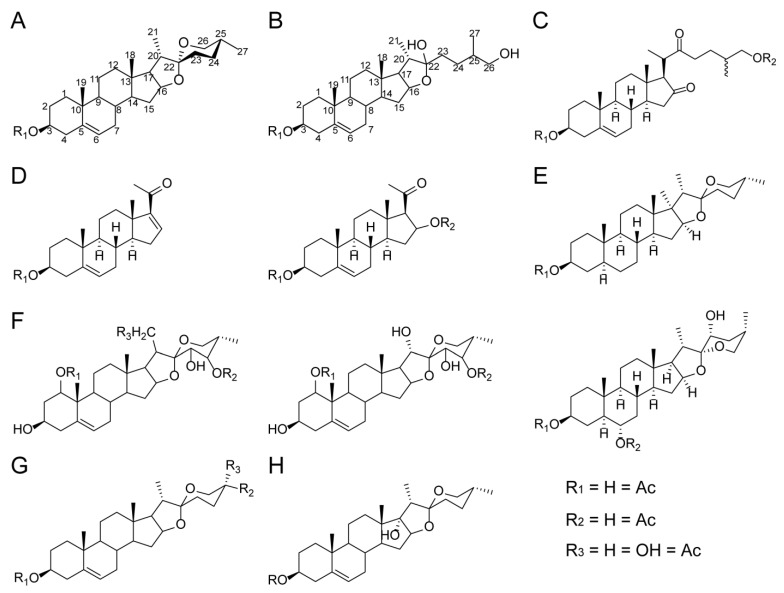
Structural diagrams of eight types of steroidal saponins isolated from plants. (**A**) Spirostanol saponins, (**B**) furostanol saponins, (**C**) cholestane saponins, (**D**) pregnane saponins, (**E**) isospirostanol saponins, (**F**) polyhydroxylated saponins, (**G**) pseudospirostanol saponins, and (**H**) pennogenin saponins. Furostanol and spirostanol saponins are the two most common types of steroidal saponins in plants, while other types of steroidal saponins are unique to different plants.

The diversity of steroidal saponins in plants is related to the differences in the aglycone structures, especially the differences in the aglycone oxidation mode, such as C-16, C-22, and C-26 sites in the aglycone skeleton of spirostanol saponins. They also occur rarely at sites such as C-1 and C-3, of spirostanol saponins; C-6, C-7, C-12, and C27 of furostanol saponins; and C-2 and C-5 of cholestane saponins. In addition, it is also possible to introduce double bonds between C-25 and C-27 of spirostanol’s aglycone skeleton, and between C-20 and C-22 of the aglycone skeleton of furostanol saponins [5,7,9]. According to the number of sugar units connected to the aglycone of steroidal saponins, they are categorized as monoglycosides, dioglycosides, trioglycosides, or tetraoglycosides, and the common sugar units are pyranose, rhamnose, and arabinopyranose [5,6,7,8,9]. The glycosylation sites of different types of steroidal saponins are different. Spirostanol saponins are usually in 1-OH, 3-OH, and 24-OH, and furostanol saponins are usually in 1-OH, 3-OH, and 26-OH [5,6,7,8,9].

Steroidal saponins have a wide range of functions and pharmacological activities due to their connections with different aglycone skeletons and different numbers of sugar chains, and there are also many different types of steroidal saponins in the same plant. Therefore, the distribution and existing forms of steroidal saponins in plants are diverse, and may also be related to the different functions of steroidal saponins in different plants. For plants, steroidal saponins can help them to cope with biological and abiotic stresses such as pathogens, herbivores, and drought [14]. Moses et al. believed that saponins in plants are produced in response to biological stress [1]. In stress reactions, such as when they are being attacked by herbivorous insects or during pathogen infection, the increase in the saponin contents in plants is usually mediated by hormones, such as jasmonic acid and salicylic acid [14,15,16]. For human beings, steroidal saponins isolated from plants have a variety of pharmacological uses, including antibacterial activity, antitumor activity, anti-inflammatory activity, and hypoglycemic and lipid-lowering effects, so they are an important industrial raw material for the production of steroidal drugs [1,3].

Due to the complexity of the chemical structure of steroidal saponins, their biosynthesis in plants and the formation and evolution of their synthetic pathways have not been thoroughly analyzed. Here, we introduce the biosynthesis, regulation, functions, and distribution of steroidal saponins in plants. At present, the research on steroidal saponin biosynthesis is focusing on the key enzymes involved in the side-chain hydroxylation, oxidation, and glycosylation of sterols (cholesterol or β-sitosterol). Although great progress has been made in the research of steroidal saponin biosynthesis, there may still be some unknown key steps, and their regulatory mechanisms are still unclear. The distribution, possible formation, and molecular evolution of steroidal saponins in plants are also reviewed here. Fully understanding the biosynthesis and molecular evolution of steroidal saponins can not only enhance our understanding of plants against biological and abiotic stresses, but also help promote the industrial production and pharmacological application of steroidal saponins.

## 2. Biosynthesis of Steroidal Saponins

Steroidal saponins are spirostanol or furostanol derivatives biosynthesized through a series of processes on the squalene molecular backbone, including oxidation, hydroxylation, and glycosylation [1,3,17]. Thus, the biosynthesis of steroidal saponins can be divided into three stages: the biosynthesis of 2,3-oxidosqualene via the mevalonate (MVA) pathway and 2-C-methyl-d-erythritol-4-phosphate (MEP) pathway, cholesterol/β-sitosterol generated by a series of catalytic reactions of 2,3-oxidosqualene, and steroidal saponins formed by the hydroxylation, oxidation, and glycosylation of cholesterol or β-sitosterol at side chains, such as the C-16, C-22, and C-26 positions [1,3,17,18,19] (Figure 2).

### 2.1. Biosynthesis of 2,3-Oxysqualene

The precursor of 2,3-oxsqualene is isopentenyl diphosphate (IPP). There are two ways to synthesize IPP in plants, namely MVA and MEP (Figure 2). The MVA pathway is found in the cytoplasm and mitochondria. Acetyl Co-A is catalyzed by acetyl Co-A acetyltransferase (ACAT) to form acetoacetyl Co-A, which is then catalyzed by 3-hydroxy-3-methylglutaryl-CoA synthase (HMGS), mevalonate kinase (MVA), phosphomevalonate kinase (PMK), and mevalonate-5-diphosphate decarboxylase (MVD) to generate IPP (isopentenyl-5-diphosphate) [1,3,20].

The MEP pathway is found in plastids. Glyceraldehyde-3-phosphate and pyruvate form 1-deoxy-D-xylulose 5-phosphate (DXP) under the catalysis of 1-deoxy-D-xylulose 5-phosphate synthase (DXS). Then, DXP is catalyzed by 1-deoxy-D-xylulose 5-phosphate reductoisomerase (DXR), 2-C-methyl-D-erythritol 4-phosphate cytidylyltransferase (MCT), 4-diphosphocytidyl-2-C-methyl-D-erythritol kinase (CMK), 2-C-methyl-D-erythritol 2,4-cyclodiphosphate synthase (MDS), 4-hydroxy-3-methylbut-2-enyl pyrophosphate (HMBPP) synthase (HDS), and HMBPP reductase (HDR) to generate IPP (isopentenyl -5-diphosphate) [1,3,20].

Subsequently, IPP is catalyzed by isopentenyl diphosphate isomerase (IDI), geranyl pyrophosphate synthase (GPPS), and farnesyl pyrophosphate synthase (FPPS) to synthesize farnesyl pyrophosphate (FPP). Finally, FPP is condensed by squalene synthase (SQS) to form squalene, which is further epoxidized to 2,3-oxidosqualene by squalene epoxidase (SQE) [1,3,20].

### 2.2. Biosynthesis of Cholesterol/β-Sitosterol

2,3-oxidosqualene is usually cyclized from a linear to a polycyclic structure by various oxidosqualene cyclases (OSCs), and the cyclization of 2,3-oxidosqualene also marks the demarcation point between sterol and triterpene metabolism in plants. 2,3-oxidosqualene is cyclized by cycloartenol synthase (CAS) to generate the plant sterol precursor cycloartenol [18]. Cholesterol/β-sitosterol are synthesized from cycloartenol under the catalysis of a series of enzymes, such as sterol side-chain reductase 1/2 (SSR1/2), sterol C-24 methyltransferase (SMT), C-4 sterol methyl oxidase (SMO), cyclopropylsterol isomerase (CPI), sterol C-14 demethylase (CYP51), sterol C-14 reductase (C14-R), sterol 8,7 isomerase (8,7 SI), sterol 5(6) desaturase (C5-SD), and 7-dehydrocholesterol reductase (7-DR) [18,20,21,22] (Figure 2).

### 2.3. Modification of Cholesterol or Sitosterol Side Chain to Form Steroidal Saponins

As steroidal saponins are generated by a series of complex catalytic reactions, such as the oxidation, hydroxylation, and glycosylation of cholesterol or the β-sitosterol molecular skeleton, the synthesis of steroidal saponins has been controversial at present [1,19,23] (Figure 2).

Some candidate genes related to the biosynthesis of the steroidal saponin precursor diosgenin (such as *PpcyP90G4*, *PpCYP94D108*, *PpCYP94D109*, *PpCYP72A616*, *TfCYP90B50*, *TfCYP72A613*, and *TfCYP82J17*) were screened from *P. polyphylla* and *Trigonella foenum graecum* (fenugreek), and the metabolite changes in transgenic *Nicotiana benthamiana* plants were verified by metabolomics [19]. Finally, the functions of two genes, *PpCYP90G4* and *TfCYP90B50*, in plants were determined: they were involved in the formation of 22S-hydroxycholesterol by catalytic side chains of cholesterol [19]. The generated 22S-hydroxycholesterol was then catalyzed by CYPs and other enzymes to finally synthesize steroidal saponins [19,22].

However, Zhou et al. showed that the *DzCYP90B71* gene was involved in cholesterol C22-R oxidation and that the *DzCYP90G6* gene was involved in catalytic C16 oxidation through the transcriptome sequencing and targeted metabolome analysis of *D. zingiberensis* [17]. They believed that the *CYP90* gene should catalyze the formation of cholesterol, which should be 22R hydroxycholesterol, rather than 22S hydroxycholesterol [17,19].

In general, diosgenin is generated from cholesterol through the catalysis of cholesterol C16/C22-monohydroxylase (CYP90B/CYP94), cholesterol 26-hydroxylase (CYP72A), and other enzymes. The first step of the biosynthesis of steroidal saponins from diosgenin is the glycosylation of 3-O-sterol glycosyltransferase (S3GT belongs to the UGT80 family), which transfers the glycosyl group to the 3-OH site of the diosgenin [24,25].

At present, there have been few studies on the late modification of steroidal saponin aglycones, such as diosgenin. In addition to *UGT80*, Song et al. found that the *UGT73CR1* gene is involved in the later modification of steroidal saponins, such as the glycosylation of steroidal saponins, and has a higher affinity for diosgenin than *UGT80* [25,26]. In addition, furostanol glycoside 26-O-β-glucosidase (F26G) can not only catalyze the conversion of furostanol saponins to spirostanol saponins, but is also involved in catalyzing diosgenin to form steroidal saponins [20,27]. Β-sitosterol may also be catalyzed by a series of enzymes to form steroidal saponins, which needs further exploration.

Although the current plant tissue culture technology is an effective and important means to explore and verify the biosynthesis of plant-specific metabolites [28,29], it remains difficult to construct the transgenic system of many medicinal plants, such as *D. zingiberensis* and *P. polyphylla*, which leads to difficulty in the verification of many biosynthetic pathway genes. Therefore, people use microorganisms to explore the biosynthetic mechanism of steroidal saponins [28,30], and the construction of engineering strains seems to be the most effective and economical method to produce natural plant-based drugs.

Cheng et al. first constructed a yeast chassis strain (DG-Cho) capable of producing cholesterol and then co-expressed the *DzinCYP90G6* and *DzinCYP94D144* (identified from *D. zingiberensis*) and *VvCPR* (cytochrome P450 reductases) genes from *Vitis vinifera* in the constructed strain. Finally, engineering strain DG002, which can convert cholesterol into diosgenin, was obtained [30]. The DG002 strain constructed by Cheng et al. significantly improved the conversion rate of cholesterol into diosgenin [30]. In addition, the constructed engineering strains can also be used to verify the function of the key genes involved in steroidal saponin biosynthesis isolated and identified from different plants.

It has also been found that, by modifying the synthetic pathway of ergosterol in microorganisms, they can synthesize cholesterol [31,32]. In addition to using cholesterol to synthesize steroidal saponins, microorganisms can also use their own enzyme system to convert steroidal saponins, thus increasing the types of steroidal saponins. Pgas-1 isolated and purified from *Aspergillus oryzae* can hydrolyze various sugars at different positions of protodioscin, thus producing various types of steroidal saponins [33]. It has been found that *Absidia coerulea* is able to convert the steroidal saponins in *D. zingiberensis* to produce five new steroidal saponins [34]. The endophytic fungus (strain 39) isolated from *Dioscorea nipponica* increased the content of steroidal saponins in the ethanol extract of the rhizome in vitro [35]. The production of steroidal saponins in *P. polyphylla* can be promoted by using *Cylindrocarpon* sp., *Scutellospora calospora*, and *Trichoderma viride* [36,37].

## 3. Molecular Regulation of Steroidal Saponin Biosynthesis in Plants

At present, there have been few in-depth studies on the regulation mechanism of steroidal saponin biosynthesis in plants. Steroidal saponins may be induced by a variety of elicitors, including abiotic and biotic elicitors. Methyl jasmonate (MeJA) is used as an abiotic elicitor to induce the synthesis of steroidal saponins in fenugreek and *D. zingiberensis* [38,39]. Ethylene, salt stress, and melatonin had the same effect as MeJA on increasing the production of steroidal saponins [40,41]. Biotic elicitors usually include fungi or bacteria; for example, *Aspergillus niger*, *Fusarium oxysporum* Dzf17, and *Saccharomyces cerevisiae* are used as fungal elicitors to increase diosgenin production in *Helicteres isora*, while *Escherichia coli* and *Bacillus subtilis* are used as bacterial elicitors to significantly increase diosgenin production [42]. However, the specific molecular regulation mechanism of steroidal saponin biosynthesis is still unknown. One of the main reasons is that it is difficult to establish an appropriate and efficient transgenic system in medicinal plants (such as *D. zingiberensis*, *P. polyphylla*, and fenugreek), which hinders the verification of gene functions and makes it difficult to conduct in-depth research.

Moses et al. believed that the molecular regulation mechanism of steroidal saponin biosynthesis in plants is multipronged, including environmental factors and various regulatory proteins, such as transcription factors (TFs), protein kinases, and phosphatases [1]. Among them, transcription factors are considered to play an important role in the regulation of the biosynthesis of steroidal saponins, steroidal glycoside alkaloids, and triterpene saponins [1,43,44,45].

Steroidal glycoside alkaloids (SGAs) and steroidal saponins have the same biosynthetic source; that is, they both use cholesterol as a synthetic precursor [1]. Therefore, the regulation of SGA biosynthesis can be used as a reference for research on the regulation of steroidal saponin biosynthesis. It has been reported that SGAs can also be regulated by multiple transcription factors. For example, *GLYCOALKALOID METABOLISM 9* (*GAME9*) is an AP2/ERF transcription factor that can positively regulate the biosynthesis of SGAs in tomato and potato [46]. Specifically, *GAME9* can change the expression levels of the upstream MVA pathway genes of SGA biosynthesis and cholesterol synthesis gene *SSR2*, and cholesterol synthesis gene *C5-SD* is the direct target of *GAME9*. Moreover, *GAME9* plays a regulatory role in SGA synthesis by cooperating with the *SlMYC2* transcription factor [18,46,47].

The *GAME* genes seem to be the target genes of the SGA biosynthetic pathway, and the silencing of the *GAME11* gene leads to decreased levels of α-lycoposide in tomato [48]. In addition, MYB, AP2/EREBP, bHLH, WRKY, and other TFs may be involved in the biosynthesis of SGAs in *Fritillaria imperialis* and *Fritillaria anhuiensis* [43,44,49]. The jasmonate-responsive transcription factors of the ETHYLENE RESPONSE FACTOR (ERF) family (JREs) can regulate SGA biosynthesis by binding to the GCC box-like promoter elements of SGA biosynthesis genes in tomato [47].

Phytosterols are important metabolites of steroids, which include cholesterol, β-sitosterol, campesterol, and so on. Campesterol is a precursor of brassinosteroid (BR), which is a competitive synthesis compound of steroidal saponins and cholesterol or β-sitosterol, as a precursor of steroidal saponins [20,50]. The structure of BR is close to the molecular skeleton structure of steroidal saponins; moreover, we have found that BR is also regulated by drought and salt stress [25,51,52]. Therefore, the research results of BR biosynthesis regulation can also be used as a reference to explore the molecular regulation of steroidal saponins in plants.

The BRI1-EMS-SUPPRESSOR1 (BES1) and BRASSINAZOLE-RESISTANT1 (BZR1) families are the core TFs [53,54]; they can not only regulate BR synthesis, but also interact with other TFs to regulate BR synthesis. RD26 is an NAC transcription factor induced by drought and inhibiting the BR response [52]. WRKY46, WRKY54, and WRKY70, which are WRKY transcription factors, have the same effect as RD26 [51], and MYB TFs have been reported to regulate BR; for example, *GmMYB14* overexpression decreases the BR content in soybean [55]. *DWF4*, which is a C-22 hydroxylase, is a target gene downstream of the regulatory pathway of BR biosynthesis. The PIF4-BES1 complex activates the promoter of the *DWF4* gene to increase the accumulation of BRs and serve as a rate-limiting step of the BR biosynthetic pathway in *Arabidopsis* [53,56].

## 4. Functions of Steroidal Saponins in Plants

Steroidal saponins distribute and accumulate in tissue or organ-specific ways at different stages of plant growth and development, indicating that steroidal saponins may also play a role in plant growth and development [3,14,57,58]. The ability of plants to synthesize specific metabolites is essential for their survival. As far as steroidal saponins are concerned, the content and composition of steroidal saponins in plants are affected by the environment and growth stage [14].

In plants, steroidal saponins, like other plant antitoxins, act as a line of defense against infection by pathogens. The difference is that steroidal saponins are mainly accumulated and stored in plants and the biosynthesis and accumulation of steroidal saponins require a lot of energy, but plant antitoxins are more often produced by stress, so energy consumption is small [58]. However, there is another possibility: when plants are infected by pathogens, the content of precursor substances for steroidal saponin synthesis will increase due to stimulation, which will also lead to an increase in the steroidal saponin content, thus reducing energy consumption, to a certain extent [59]. Take the research on the relationship between saponins and pathogens in oat (*Avena sativa*) as an example: both triterpenoid saponins and steroidal saponins are present in oat, and the triterpenoid saponins (avenacins) are accumulated in root tip epidermal cells through active glycosylation, while avenacins have strong antifungal activity and can be released into the soil rhizosphere [60,61,62]. Compared with triterpenoid saponins, steroidal saponins in oats are stored in the vacuoles of plants as inactive disaccharides. When pathogenic bacteria infect plant tissues and destroy cell membranes, steroidal saponins will be activated so that β-Glucosidase hydrolyzes the D-glucose unit to form toxic single anthocyanin, which destroys the fungal plasma membrane by forming membrane pores, thus leading to fungal cell death [63,64].

Another role of steroidal saponins in plants is to protect them from phytophagous animals or insects. The main mechanism of action is to inhibit feeding by herbivores and insects as toxins and digestive inhibitors [65,66,67]. However, the specific mechanism may be that steroidal saponins have cytotoxicity and can destroy the cell membranes of blood cells and many animal cells [68]. Another possible explanation is that steroidal saponins lead to the penetration of insect cell membranes and the loss of cell integrity [58]. It has also been reported that steroidal saponins isolated from *Allium porrum* have a strong antifeedant effect on Lepidoptera insects, such as *Peridroma saucia* and *Mamestra configurata* [69].

## 5. Distribution of Steroidal Saponins in Plants

Steroidal saponins are synthesized and distributed in different plants. According to the current research, they are mainly distributed in monocots, such as the Dioscoreaceae (*Dioscorea* species), Melanthiaceae (*Paris* species), Smilacaceae (*Smilax* species), Asparagaceae (*Asparagus* species), Poaceae (*Avena sativa*), and Arecaceae (*Phoenix dactylifera*) [1,3]. Steroidal saponins are also distributed in a few eudicots, such as the Solanaceae (*Solanum* species) and Fabaceae (fenugreek) [1,3].

However, the composition of steroidal saponins in different plants varies greatly. For example, the main saponins that accumulate in *Dioscorea* are the five-ring furostanol saponins (parvifloside, protodioscin, and protobioside) and the six-ring spirostanol saponins (dioscin and gracillin). In addition to the accumulation of the above two steroidal saponins, four-ring pregnane saponins (such as parisyunnanoside J) and six-ring polyhydroxylated saponins (such as parisyunnanosides G/H/I and padelaoside B) accumulate in Paris plants [8,70] (Figure 1, Appendix A). In addition, the organs or tissues in which steroidal saponins accumulate in different plants are also different; for example, *Dioscorea* plants accumulate steroidal saponins in underground rhizomes (*D. zingiberensis*) or tubers (*D. composita*) and fenugreek accumulates them in its seeds [70,71,72,73].

### 5.1. Arecales Species

At present, there has been very little research on steroidal saponins in Arecaes plants, and the research that does exist mainly focuses on *Phoenix dactylifera* [74]. Based on ESI-MS analysis, Hamed et al. identified 21 steroidal saponins from methanol extracts of *P. dactylifera* pollen, among which Protodioscin was the most common. Two newly identified steroidal saponins were also reported: 3-o-histi-dine-26-o-hexosyl-dioscin and 3-o-histidine-26-o-dihexosyl hydroxydioscin [74] (Appendix A).

### 5.2. Asparagales Species

In contrast, Asparagales plants contain relatively more steroidal saponins, including the Amaryllidaceae, Agavaceae, Lomandroideae, Nolinoideae, Asparagoideae, and Asphodelaceae [10,75,76,77,78]. In addition, the type and distribution of steroidal saponins in these plants are also different.

At present, the main plant species containing steroidal saponins in the Amaryllidaceae include the Allioideae, *Allium* species. According to the structure of aglycone, steroidal saponins isolated from the flowers, bulbs, seeds, and underground parts of *Allium* plants can be divided into three types: spirostanol saponins with diosgenin or agigenin as aglycones, furostanol saponins with sugar chains at the C-3 and C-26 positions, and cholestane saponins with C-5 (C-6) unsaturated double bonds [10,79,80,81,82] (Figure 1, Appendix A). The sugar residues linked to steroidal saponins in *Allium* plants are mainly monosaccharides or disaccharides, but there are a few trisaccharides [10].

Compared with the Amaryllidaceae, *Agave* species are the main plants containing steroidal saponins in the Agavaceae, and a small number of steroidal saponins have also been identified from a few Agavaceae plants, such as *Camascia* and *Yucca* species [83,84,85]. The types of steroidal saponins isolated from Agavaceae plants are mainly furostanol saponins and spirosterol saponins. Meanwhile, steroidal saponins are mainly distributed in the leaves, fruits, flowers, and rhizomes of Agavaceae plants [75,86,87,88] (Appendix A). Spirostanol saponins can be divided into monosaccharides, disaccharides, and polysaccharides according to the linked glycosyl units, while furostanol saponins are mainly disaccharides and a few trisaccharides [75,86,87,88].

As for Lomandroideae plants, only five steroidal saponins have been identified in *Cordyline stricta* [89]. Among the Nolinoideae plants, the plants containing steroidal saponins are mainly *Polygonatum*, *Dracaena*, and *Sansevieria* species [9,64]. About 180 different saponins were isolated from *Dracaena* and *Sansevieria* plants, mainly furostanol saponins and spirostanol saponins, as well as a small number of cholestane saponins, pregnane saponins, and β- Sitosterol saponins [9]. In traditional Chinese medicine, the rhizomes of some *Polygonatum* plants are also known as “Yuzhu”, which are used to treat osteoporosis, diabetes, and lung disease [64,77,78]. Steroidal saponins isolated from the rhizomes of *Polygonatum* plants can be divided into three types: cholestanol-type (such as polygonatumoside A/B/C), furostanol-type (such as polygodoside G), and spirostanol-type (such as polygonoside A/B/C) [77,90,91] (Appendix A).

Among the Asparagoideae plants, steroidal saponins are mainly distributed in Asparagus species [92]. Not many steroidal saponins have been isolated from the roots and fruits of Asparagus plants, and the most common one is protodioscin [93,94]. Only a small number of steroidal saponins were isolated from the aerial parts of *Hemerocallis fulva* var. *kwanso* in the Asphodelaceae [78] (Appendix A).

### 5.3. Dioscoreales Plants

Dioscoreales plants are an ancient branch of monocots, among which Dioscoreaceae and Taccaceae contain steroidal saponins. Among the Dioscoreaceae plants, only *Dioscorea* plants contain steroidal saponins. It has been reported that, among the over 600 species of *Dioscorea*, about 137 contain steroidal saponins, of which 41 species of *Dioscorea* contain more than 1% diosgenin [95]. More than 190 steroidal saponins have been isolated and identified from *Dioscorea* plants, including spirostanol saponins and furostanol saponins. These steroidal saponins have also been used as chemical taxonomic markers of *Dioscorea* plants [70,96].

Among the nine major clades of *Dioscorea* pants, steroidal saponins were mainly distributed in the early diverged lineage of *Dioscorea* plants, such as *Dioscorea* sect. *Stenophora* [20]. The common plants containing steroidal saponins in *Dioscorea* sect. *Stenophora* are *D. zingiberensis*, *D. panthaica*, *D. parviflora*, *D. futschauensis*, *D. septemloba*, *D. nipponica*, and *D. collettii* var. *hypoglauca* [70,72,97]. More than 70 steroidal saponins have been isolated from the rhizomes of *Dioscorea* sect. *Stenophora* plants, mainly including the following two types: spirostanol saponins and furostanol saponins, and the content of spirostanol saponins is higher than that of furostanol saponins [72]. The most common of these steroidal saponins include dioscin, gracillin, protogracillin, parvifloside, protodeltonin, trillin, methyl protogracillin, and protobioside [70,72]. Species-specific steroidal saponins include progenin II and III and dioscoreside C/D/E in *D. panthaica* [97], and spongipregnoloside A-D and spongioside A/B in *D. septemloba* [98]. A few pregnane saponins have also been isolated from *D. spongiosa* [5,99] (Appendix A).

Other clades of Dioscorea species, such as *Dioscorea* sect. *Combilium* and *Dioscorea* sect. *Opsophyton*, also contain a few steroidal saponins, including *D. esculenta* and *D. bulbifera* [100,101]. However, there has been little research on steroidal saponins in the *Dioscorea* sect. *Shannicorea*, *Dioscorea* sect. *Lasiophyton*, *Dioscorea* sect. *Botryosicyos*, and *Dioscorea* sect. *Enantiopheyllum* plants are mainly distributed in China. Among the *Dioscorea* plants distributed outside China, *D. composita*, native to Mexico, also contains a high content of steroidal saponins, which is another important steroidal drug resource plant, in addition to *D. zingiberensis* [95]. It has been determined that the content of diosgenin in the tubers of *D. composita* is 3.68 ± 0.5% (dry weight) [94]. In Taccaceae, only *Tacca chantrieri* contains steroid saponins; about five furostanol saponins and two spirostanol saponins were identified from its rhizome [102] (Appendix A).

### 5.4. Liliales Plants

Liliales is an early diverged lineage of monocots, and the plants in the lineage containing steroidal saponins are the Liliaceae, Melanthiaceae, and Smilacaceae.

(1)Liliaceae plants. At present, the research on steroidal saponins of Liliaceae plants has mainly focused on *Lilium* and *Fritillaria* plants [103]. More than 80 steroidal saponins have been isolated and identified from the bulbs of *Lilium lancifolium*, *L pumilum*, *L. longiflorum*, *L. candidum*, *L. speciosum*, *L. tenuifolium*, *L. callosum*, and other *Lilium* plants, which are mainly divided into four types: spirostanol-type, furostanol-type, isospirostanol-type, and pseudospirostanol saponins [103]. Specifically, spirostanol saponins isolated from *Lilium* plants mainly include lilioglycoside B/H and brownioside, which are unique to *Lilium* plants [103,104,105,106,107]; the isolated isospirostanol saponins include lilioglycoside C, D, and I, respectively; and furostanol saponins mainly include lilioglycoside K, N, R and pardarinoside A–D, F, and G [103,104,105,106,107]. Among *Fritillaria* plants, only 12 steroidal saponins have been isolated and identified from the bulbs of *Fritillaria pallidiflora*, such as pallidiflosides D/E/G/H/I, protobioside, and Polyphyllin V [108] (Appendix A).(2)Melanthiaceae plants. Among the plants in this family, *Paris* and *Trillium* are the largest two genera; they are important Chinese medicine plants with a long medicinal history [109,110]. A variety of steroidal saponins have been isolated and identified from the rhizomes of Paris plants, including spirostanol-type, furostanol-type, cholestane-type, pregnane-type, and polyhydroxylated-type saponins [8]. Steroidal saponins isolated and identified from Trillium plants can be divided into the following types: spirostanol, furostanol, and pennogenin saponins. Among them, the specific saponins of Trillium plants are trillins, trikamsteroside C, trillenoside A, parisapioside C, etc. [111] (Appendix A).(3)Smilacaceae plants. The most common plants containing steroidal saponins are *Smilax* plants [7]. A total of 104 steroidal saponins have been isolated from about 20 species of *Smilax* plants [7]. These steroidal saponins include five types: spirostanol-type, isospirostanol-type, furostanol-type, pregnane-type, and cholestane-type; most of the linked sugar groups are monosaccharides or disaccharides [7,112,113,114,115].

Steroidal saponins are also found in a few monocots, such as Poales and Zingiberales. Poaceae are only *Avena* plants (oats), such as *Avena sativa* and *A. strigosa*. A total of 16 steroidal saponins were isolated and identified from oats, including Avenacoside A, B, C, D, etc. [116]. Oats are also one of the few plants that contain both triterpenoid saponins and steroidal saponins. However, *Costus Spiralis* and *C. speciosus* are the only Zingiberales containing steroidal saponins in their rhizomes [117,118] (Appendix A).

### 5.5. Eudicots

In eudicots, steroidal saponins were identified mainly from a few plants, such as the Fabales, Ranunculales, and Solanales. Among them, Fabaceae plants only contained a small number of steroidal saponins in the seeds of fenugreek [71,73]. A small number of steroidal saponins was identified in two plants (*Helleborus niger* and *H. orientalis*) in the Ranunculaceae, including one polyhydroxy saponin: polyhydroxy hellebosaponins [118,119]. In Solanaceae plants, steroidal saponins are mainly distributed in Solanum plants. For example, three furostanol saponins, four spirostanol saponins, two cholestane-type saponins, and one spirostanol saponin were identified in *Solanum abutilides*, *S. anguivi,* and *S. melongena* [120,121,122,123] (Appendix A).

In addition, more than 17 steroidal saponins, including Terrestrinin C-T, were identified in the Zygophyllales plant *Tribulus terrestris* [124,125].

It can be seen from the above research results that steroidal saponins are mainly distributed in the early diverged lineages of monocots, including the Asparaceae (*Asparagus*), Dioscoreaceae (*Dioscorea*), Liliaceae (*Lilium*), Melanthiaceae (*Paris* and *Trillium*), and Smilacaceae (*Smilax*). There are also a few eudicots, mainly distributed in Solanaceae plants, that produce these compounds (Appendix A). Furthermore, Li et al. evaluated the evolution and distribution pattern of steroidal saponins in *Dioscorea* plants through ancestral states analysis [20]. Their results showed that steroidal saponins were distributed intensively in *Dioscorea* sect. *Stenophora* species. With the evolution of *Dioscorea* species, the frequency of steroidal saponins in plants is decreasing, with only small numbers in the more evolved clades, such as New world I, New world II, sect. *Opsophython*, and Malagasy clade [20,126]. A large number of ancestral nodes of steroidal saponins have been evaluated in the *Dioscorea* sect. *Stenophora* branches, indicating that steroidal saponins are an ancestral trait in *Dioscorea* species, mainly distributed in the early differentiated lineages of *Dioscorea*, and are phylogenetically conserved [120].

However, the mechanism of the distribution and accumulation of steroidal saponins in a few species of the above families and genera and how the distribution differences evolved are still unknown. The answers to these questions are of great significance for the systematic understanding of the distribution and evolution pattern of steroidal saponins in plants.

## 6. Molecular Evolution of Steroidal Saponin Biosynthetic Pathway Genes

Steroidal saponins are mainly distributed in *Dioscorea*, *Paris*, *Asparagus,* and other plants. At present, only *D. zingiberensis*, *Dioscorea rotundata*, asparagus, *P. dactylifera*, oats, and other plant genomes have been assembled and annotated. However, steroidal saponins are only abundant in *D. zingiberensis*, and the content is relatively small in other plants. Moreover, the genomes of most plants containing steroidal saponins, such as *Allium*, *Paris*, *Smilax*, *Trillium*, and *Polygonatum*, have not yet been reported. In the absence of species genome data, only comparative transcriptome analysis can be used to analyze the biosynthesis and evolution of plant steroidal saponins to a limited extent.

Therefore, it is the first/best choice to analyze the biosynthesis and evolution of steroidal saponins by sequencing and assembling the genome of *D. zingiberensis*. Other plants, such as fenugreek and trillium, are also suitable research materials. Li et al. assembled a high-quality chromosome-level genome of *D. zingiberensis* by combining Illumina and Oxford Nanopore technologies [20]. Through comparative genomic analysis, it was found that there were a large number of species-specific genes in the genome of *D. zingiberensis*, and these specific genes are significantly enriched in the biological processes of the responses to pathogens, oxidative stress, etc. This finding indicates that *D. zingiberensis* has evolved a large number of specific genes for coping with stress in order to adapt to the environment, and there are many *D. zingiberensis*-specific genes involved in the biosynthesis of steroidal saponins [20]. These results show that the genome of *D. zingiberensis* produced a relatively large number of genes involved in steroidal saponin biosynthesis during the evolutionary process, which may also be the premise or basis for steroidal saponin synthesis in *D. zingiberensis* [20].

Furthermore, there are a large number of expanded gene families in the *D. zingiberensis* genome, and these gene families are significantly enriched in UDP-glycosyltransferases activity, mono-oxygenase activity, steroid biosynthesis, stress response, and other processes related to steroidal saponin synthesis [20]. Moreover, these expanded gene families also include the gene family members of *OSC*, *CYP450*, and *UGT*, which play important roles in several key steps of steroidal saponin biosynthesis [1,20]. Thus, compared with other plants, the species-specific genes and significantly expanded gene families in the genome of *D. zingiberensis* provide abundant genetic and evolutionary resources related to steroidal saponin biosynthesis [20].

The expansion or contraction of gene families is an important factor in species differentiation, phenotypic diversification, and adaptation to natural changes [127,128]. Similarly, whole-genome duplication (WGD) events are also considered to be the major driving force of species diversity and phenotypic variation [129,130,131]. Specifically, the role of WGD and gene tandem duplication is mainly to produce a large number of gene copies in the genome. Further analysis showed that a large number of copies of key genes involved in the biosynthesis of steroidal saponins in *D. zingiberensis*, including *CAS*, *CYP72A*, *CYP90B*, *CYP94*, *S3GT*, and *F26G*, were produced through WGD and gene tandem duplication events [20]. Combined with the above results, gene tandem duplication coupled with WGD events in the genome may be the driving force to promote the evolution of the steroidal saponin biosynthetic pathway in *D. zingiberensis* [20].

Gene replication can help plants to obtain more gene copies, which may produce new gene functions that are more adaptive than single-copy or low-copy genes [130]. For example, sterol side-chain catalytic enzyme-encoding genes, such as *CYP90B*, *CYP94*, and *CYP72A*, play an important role in steroidal saponin biosynthesis, and the functions of these genes vary between different plants [5,18,132]. In plants without steroidal saponins, the *CYP90* gene is involved in catalyzing the conversion of cholesterol into 22S hydroxycholesterol and then into brassinosteroids. However, in plants containing steroidal saponins, such as *D. zingiberensis*, *P. polyphylla*, and fenugreek, the new *CYP90* gene generated through gene duplication has evolved new functions, such as sterol 16,22-polyhydroxylation during molecular evolution, which can catalyze the synthesis of cholesterol into diosgenin [19,20,133]. This also indicates that diosgenin and steroidal saponins, as new defense compounds, coordinate plant growth and defense in the evolution of plants containing steroidal saponins. Similarly, the main function of the *CYP72A* gene in plants is involved in the 13 hydroxylation of gibberellin, while in plants containing steroidal saponins, it is involved in the biosynthesis of steroidal saponins [19,38,134,135].

Duplicated genes usually have newly evolved functions that drive the evolution of the biosynthetic pathways of specific metabolites [79,136]. It can be inferred that the new members of the CYP450 and UGT gene families (such as *CYP90*, *CYP94*, *CYP72A*, *UGT73*, and *UGT80* genes) generated by the gene tandem duplication and WGD event in the *D. zingiberensis* genome provide abundant genetic and evolutionary resources for the formation and evolution of the steroidal saponin biosynthetic pathway. Christ et al. also found that the *CYP90* gene from *P. polyphylla* and fenugreek was mainly derived from gene duplication events in plants [19].

Phylogenetic analysis shows that members of the UGT80 family have been reported in algae, bryophytes, ferns, and angiosperms, indicating that UGT80 is the ancestor of the UGT family in plants [24,26]. Song et al. believe that the UGT73 family appeared after the emergence of ferns; however, a large number of members appear in Arabidopsis, *O. sativa*, and *P. polyphylla* [24,26,137]. The UGT73 family members in plants contain a variety of functional enzymes involved in the glycosylation of plant secondary metabolites, including flavonoids, triterpenes, steroidal saponins, and alkaloids [138,139]. In contrast, UGT80 has more diverse functions in plants, and is involved in the catalysis of sterols, steroidal glycoside alkaloids (SGAs), and steroidal saponins [25,26,140].

In conclusion, we speculate that the gene duplication events in the plant genome may provide the driving force for the species-specific evolution of steroidal saponin biosynthesis in plants; further, new defense substances, such as diosgenin and steroidal saponins, evolve to help plants coordinate their growth and defend against stress. In our opinion, it is necessary to verify the function of steroidal saponin biosynthesis genes in plants containing steroidal saponins using transgenic methods. It should also be noted that the current speculation needs further systematic exploration and verification in combination with phylogenetic analysis methods.

## 7. Conclusions and Perspectives

In this review, we systematically introduced the genes involved in the biosynthesis of steroidal saponins and their functions in plants. The key biosynthetic steps mainly focus on the biosynthesis of cholesterol and the hydroxylation, oxidation, and glycosylation of its side chains. The biosynthetic pathway of cholesterol has been thoroughly studied, but the specific catalytic step (i.e., the modification of the cholesterol C22 position) of diosgenin from cholesterol is still controversial. In addition, the production of different steroidal saponins by diosgenin under the catalysis of CYP450 and UGT enzymes is still under-studied. Therefore, it is necessary to further explore how diosgenin can synthesize steroidal saponins through glycosylation and other modifications by CYP450 and UGT. Moreover, there have been few in-depth studies on the regulation mechanism of steroidal saponin biosynthesis, and the identification of these various factors involved in the regulation of biosynthesis, including exogenous hormones, and transcription factors, can be used to enhance the biosynthesis of steroidal saponins in plants.

Due to the complex genetic background and metabolic composition of plants containing steroidal saponins, it is difficult to obtain plant calluses that can be used for transgenic verification, which is the greatest difficulty in the functional verification of genes related to steroidal saponin biosynthesis in plants. In addition to improving the content of steroidal saponins in plants using transgenic technology, it is also possible to use corresponding substrates in microbial systems (such as yeast) for the exogenous synthesis of steroidal saponins through synthetic biology methods. This strategy can also be used for the screening and identification of key genes for steroidal saponin biosynthesis. Therefore, the studies on steroidal saponin biosynthesis in the engineering bacteria system can provide important clues for screening and identifying new candidate genes that may also be applied to the industrial production of drugs.

We also reviewed the functions and distribution of steroidal saponins in plants and discussed the molecular evolutionary mechanism of steroidal saponin biosynthesis. These research advances can not only provide a reference for studying how steroidal saponins distribute and evolve in plants, but can also provide clues for exploring and mining new steroidal saponin resource plants.

In our opinion, without breaking through the technical problem of using transgenic plants to verify the function of pathway genes, the most effective research strategy to explore the biosynthesis and regulation mechanism of steroidal saponins is the use of microorganisms. At the same time, understanding the molecular evolution mechanism of steroidal saponins can also provide a reference for other specialized plant metabolites.

## Figures and Tables

**Figure 2 ijms-24-02620-f002:**
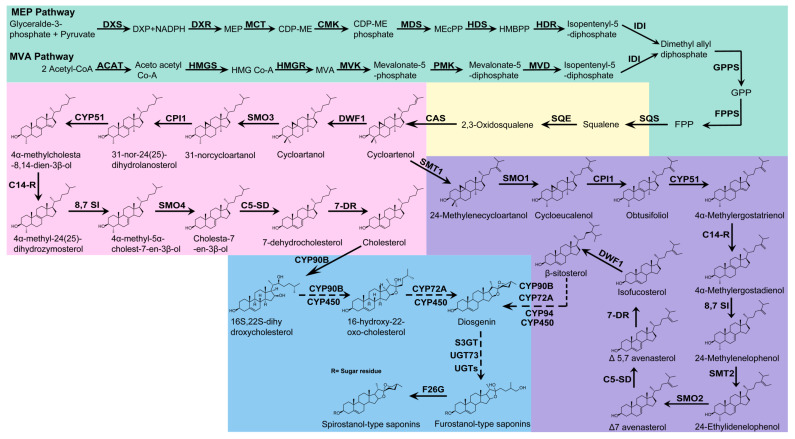
Possible biosynthetic pathways of steroidal saponins in plants. Biosynthetic pathway of 2,3-oxysqualene (light green); squalene is catalyzed by SQE to form 2, 3-oxide squalene (light yellow), which is also the demarcation point between sterol and triterpene metabolism. The biosynthetic pathway of cholesterol and β-sitosterol are depicted in pink and purple, respectively. The biosynthetic pathway of steroidal saponins is marked in blue. Dashed arrows indicate that there may be multiple catalytic steps.

## Data Availability

Not applicable.

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
