# Peer review of "Advances in the Biosynthesis and Molecular Evolution of Steroidal Saponins in Plants"

_ijms, 2023, doi:10.3390/ijms24032620_

Round 1

Reviewer 1 Report

The authors of the review entitled "Advances in biosynthesis and molecular evolution of steroidal saponins in plants" is a good compilation of the steroidal saponins, however, from my point of view the title is creating some incongruencies, I am not seeing the "molecular evolution" discussion. Raising several questions, for example which genes has been change with the time in a particular family, how the evolution plays here? I am not really sure that the title is the appropriate here. In the Biosynthesis section, the authors compiled it, but they missed and important review about it (PMID: 34617240). Seems to me that there are not much new things related to biosynthesis. Which is new in this review? The figure 2 must be in landscape orientation. After that, the authors just introduced by families the steroidal saponins, again they missed several important reviews about it, for example (PMID: 33805482). As a suggestion if the authors really want to explain the saponins' molecular evolution in plants why better they focus initially on the oldest families and the genes were evolving.  

The authors must work really hard to make more comprehensive review, the review needs major revisions

Author Response

Response to Reviewer 1 Comments

Point 1: The authors of the review entitled "Advances in biosynthesis and molecular evolution of steroidal saponins in plants" is a good compilation of the steroidal saponins, however, from my point of view the title is creating some incongruencies, I am not seeing the "molecular evolution" discussion. Raising several questions, for example which genes has been change with the time in a particular family, how the evolution plays here? I am not really sure that the title is the appropriate here.

Response 1:  Thank you for your comments and suggestions. The “molecular evolution” in our manuscript is mainly focused on the reported evolution of steroidal saponin biosynthetic pathway genes in plants. In Dioscorea zingiberensis, a large number of gene copies of CAS, CYP72A, CYP90B, CYP94, S3GT and F26G genes were generated through WGD events and tandem duplication. Therefore, gene tandem duplication and WGD events in the genome may be the driving force for the evolution of the steroidal saponin biosynthetic pathway in D. zingiberensis. On this basis, a large number of gene copies have evolved new functions. Thus, evolution not only produced a large number of gene copies, but also multiple gene functions.

At present, there are few studies on molecular evolution of steroidal saponins, and the specific mechanism needs to be further explored. Therefore, our title is “Molecular evolution of steroidal saponin synthesis pathway genes”, specifically discussing the evolution of genes. In order to summarize more updated gene evolution, we added some new content in the sixth part, hoping to summarize the research progress of molecular evolution of genes.

Point 2:  In the Biosynthesis section, the authors compiled it, but they missed and important review about it (PMID: 34617240). Seems to me that there are not much new things related to biosynthesis. Which is new in this review?

Response 2:  Thank you for your comments. We have quoted the review you mentioned (PMID: 34617240) in our revised manuscript. In this review, the authors have made a good summary of the research on the key enzymes in the biosynthesis pathway of steroidal saponins, and they also focused on the research on the synthesis of steroidal saponins by using microorganisms. Our review focuses on the biosynthesis and regulation of steroidal saponins, the distribution of steroidal saponins, and the molecular evolution of steroidal saponin in plants. We also added some research progress on the biosynthesis of steroidal saponins by microorganisms in the section of “Biosynthesis of steroidal saponins” in our manuscript.

Our new content mainly focuses on the mechanism of steroidal saponin biosynthesis in plants, such as CYP90B and CYP72A participating in the side chain modification of cholesterol; Late modification of diosgenin, such as genes UGT73 and UGT80 involved in glycosylation; And the late modification of steroidal saponins, such as F26G gene involved in the transfer of furostanol saponins to spirostanol saponins. The contents we summarized were not mentioned in the previous review, and are the results of recent studies.

Point 3:  The figure 2 must be in landscape orientation. After that, the authors just introduced by families the steroidal saponins, again they missed several important reviews about it, for example (PMID: 33805482).

Response 3: Thank you for your suggestion. We have redrawn Figure 2. We have re-referenced the review you mentioned (PMID: 33805482), and added new content in the structure of steroidal saponins and the distribution of steroidal saponins in plants.

Figure 2

Point 4: As a suggestion if the authors really want to explain the saponins' molecular evolution in plants why better they focus initially on the oldest families and the genes were evolving.

Response 4:  Thank you for your suggestion. Since there are few studies on genes related to steroidal saponin biosynthesis, we can only add as much as possible about the evolution of genes involved in steroidal saponin biosynthesis, such as CYP90, CYP94, CYP72A, UGT73, UGT80 and F26G. The biosynthesis of steroidal saponin includes upstream MVA/MEP, squalene biosynthetic and sterol biosynthetic pathways, which are involved in many core and specific metabolic pathways, not just steroidal saponins. While genes such as CYP90, CYP94, CYP72A, UGT73, UGT80 and F26G are specifically involved in the biosynthesis of steroidal saponins. Therefore, we focus on the molecular evolution of these genes. An overview of the evolution of these specific genes is very helpful for understanding the molecular evolution of steroidal saponin biosynthesis in plants.

Point 5: The authors must work really hard to make more comprehensive review, the review needs major revisions.

Response 5: Thank you for your comments and suggestions. We have carefully and comprehensively revised our manuscript according to your comments.

Reviewer 2 Report

The comments see attached.

Author Response

Response to Reviewer 2 Comments

General comments to the manuscript (MS)

Point 1: It is not clear what are the main chemical peculiarities of the cholestane saponins. The authors characterize them as saponins "with C-5 (C-6) unsaturation" (Line 43), and "As for the cholestane saponins, most of the cholestane steroidal saponins ... have C-5 (C-6) unsaturation (Line 50). However, another separate group, 'furostanol saponins ... have double bonds between C-5 and C-6" (Lines 48-49). Moreover, 8 among 9 structures in Fig. 1 have double bonds between C-5 and C-6. I cannot assume that the authors make differences between "C-5 (C-6) unsaturation" and "double bonds between C-5 and C-6".

"The glycosyl group are usually one, two or more" (Line 53) is also indistinct characteristics.

Response 1: Thank you for your comments and suggestions. We are sorry about that. As for the definition and explanation of the chemical structure of steroidal saponins, we carefully reviewed the literature, revised the corresponding contents in the manuscript. Meanwhile, we also revised the wrong chemical structure of steroidal saponins in Figure 1. The glycosyl group is not the standard to judge the type of steroidal saponins, and there are many kinds of glycosyl group with complex structures. We can only summarize the glycosyl types of steroidal saponin links isolated from plants at present, including monosaccharide, disaccharide, trisaccharide, tetrasaccharide and so on. To help readers understand more clearly, we have rewritten this part. (Lines 50-97 of the revised manuscript)

Point 2:Line 52: "cholestane type steroidal saponins are usually oxidized at C-1, C-3, C-16 and C-22 positions". However, 

– the structure " cholestane saponins" in Figure 1 is not oxidized at C-1; 

– representatives of not cholestane type, but of pregnane saponins and polyhydroxylated

saponins (Fig. 1) are oxidized at C-1;

– all skeletons in Fig. 1 are oxidized at C-3 (as a necessary condition to be classified as such

steroids); 

– several types of saponins, not only of cholestane type, contain skeletons oxidized at C-16

and C-22 positions (Fig. 1). 

Response 2:  Thank you for your comments. We are sorry about that. To avoid misunderstanding, we have rearranged and revised the part of the late modification of steroidal saponins. (Lines 85-97 of the revised manuscript)

Point 3:Section title "2. Biosynthesis of steroidal saponins" (Line 87) coincides with the subsection title "2.3 Biosynthesis of steroidal saponins" (Line 136), that is not desirable (if applicable at all).

Response 3:  Thank you for pointing out this. We have revised the titles of these two parts, and the new title is “Modification of cholesterol or sitosterol side chain to form steroidal saponins”. (Line 199 of the revised manuscript))

Point 4: Table 1 "Steroidal saponins isolated from plant" seems disproportionally voluminous (eight and a half pages incorporated in twenty pages of the MS without Refs). I suppose that the table entries of the type "five ... saponins", "three ... saponins", etc., without any chemical structural definitions, should be deleted. Or the Table 1 may be a Supplementary Material. 

Response 4:  Thank you for your suggestions and comments. In order to reduce the length of the article, we have used Table 1 as a supplementary material.

Point 5:The section 7 "Conclusion and perspectives" tries to touch upon issues related to the most effective research strategy to explore the biosynthesis and regulation mechanism of steroidal saponins by using microorganisms (Line 553). This problem should be discussed in the previous sections or should not be mentioned incidentally.

Response 5:  Thank you for your comments. We have added some research progress on the biosynthesis of steroidal saponins by microorganisms in the section of “Biosynthesis of steroidal saponins” in our revised manuscript. (Lines 233-259 of revised manuscript)

Reviewer 3 Report

As mentioned below linguistic challenges and the size of Figure 2 have prevented me from understanding this manuscript. I, therefore, suggest that the manuscript has to undergo severe linguistic revision before it can be evaluated.

I miss a definition of the topics of this review steroids and saponins. In old textbooks of pharmacognosy, saponins were defined as compounds, which could hemolyze red blood cells and reduce the surface tension of aqueous solutions.

Why are the nine representatives in Figure 1 selected?

A general problem is illustrated in lines 60 to 73. The different needs of different plants. The word need indicates that the plants have chosen to make steroids. Probably evolution has given plants producing saponins some advantages enabling them to survive in contrast to other plants. Thus, saponin production probably is a stochastic developed property developed in successful species in contrast to failing species. Thus, the plants did not develop saponins to overcome stress but survived because they produced saponins.

I assume that Figure 2 is the essential figure in this manuscript revealing intermediates and products for saponin biosynthesis.  Unfortunately, Figure 2 can not be read. It must be enlarged. As a consequence of this bad figure I can not evaluate paragraph 2.3.

After huge enlarging, I realized that both the deoxyxylolose and the mevalonate pathway are mentioned as sources for squalene. Is that true? In general, triterpenoids are believed to drive from mevalonic acid.

Another problem for me is that cholesterol is mentioned as a precursor of saponins. Some textbook claim that triterpenes in plants are derived from cycloartenol. In paragraph 2.2 cycloartenol synthase is mentioned and it is stated (I guess) that cycloartenol is a precursor for cholesterol. Is this true?

Why is the steroid skeleton depicted differently in the blue and pink parts of figure 2?

The review is stated to focus on monocots. Why is Fabaceae then mentioned?

Table 1: An impressive table mentioning steroids in different plant species. In column 3 (Plant groups) Firstly I do not know what a plant group is, secondly why is the name of the genus mentioned when this also appears I column 2?

Linguistic problems:

Line 153 Diosgenin generated by cholesterol should be generated from.

Line 123 Cholesterol/β-sitosterol are synthesized by cycloartenol, and might be synthesized from cycloartenol

163 but this part of the research is controversial, but this suggested pathway is dubious.

189 it is stated that plant steroids have cholesterol as a precursor. Previously it was assumed that plant steroids have cycloartenol as a precursor.

Line 286 Amaryllidaceae plants mainly contain steroidal saponins at present, I do not understand the message of this sentence.

Line 294 Compared with Amaryllidaceae, Agave species are the main plants contain in 294 Agavaceae, and a few Agavaceae plants such as Camassia and Yucca species have also 295 identified steroidal saponins. I do not understand the message of this sentence.  

Line 447 to 456: Saponins are also found in dicots.

Line 470 plant name in italic

 As mentioned below linguistic challenges and the size of Figure 2 have preventd me form understanding this manuscript. I therefor suggest that the manuscript has to undergo severe linguistic revision before it can be evaluated.

I miss a definition of the topics of this review steroids and saponins. In old textbooks of pharmacognosy saponins were defined as compounds, which could hemolyze red blood cells and reduce surface tension of aqueous solutions.

Why are the nine representatives in Figure 1 selected?

A general problem is illustrated in line 60 to 73. The different needs of different plants. The word need indicate that the plants have chosen to make steroids. Probably evolution has given plants producing saponins some advantages enabling then to survive in contrast to other plants. Thus, saponin production probably is a stochastic developed property developed in successful species in contrast to failing species. Thus, the plants did not develop saponins to overcome stress, but survived because they produced saponins.

I assume that Figure 2 is the essential figure in this manuscript revealing intermediates and products for saponin biosynthesis.  Unfortunately, Figure 2 can not be read. It must be enlarged. As a consequence of this bad figure I can not evaluate paragraph 2.3.

After huge enlarging I realized that both the doxyxylolose and the mevalonate pathway are mentioned as sources for squalene. I that true? In general, triterpeoids are believed to drive from mevalonic acid.

Another problem for me is that cholesterol is mentioned as precursor of saponins. Some textbook claim that triterpenes in plants are derived from cycloartenol. In paragraph 2.2 cycloartenol synthase is mentioned and it is stated (I guess) that cycloartenol is a precursor for cholesterol. Is this true?

Why is the steroid skeleton depicted different in the blue and pink part of the figure 2?

The review is stated to focus on monocots. Why is Fabaceae then mentioned?

Table 1: An impressive table mentioning steroids in different plant species. In column 3 (Plant groups) Firstly I do not know what a plant group is, secondly why is the name of the genus mentioned when this also appear I column 2?

Linguistic problems:

153 Diosgenin generated by cholesterol should be generated from.

123 Cholesterol/β-sitosterol are synthesized by cycloartenol, might be synthesized from cycloartenol

163 but this part of the research is controversial, but this suggested pathway is dubious.

189 it is stated that plant steroids have cholesterol as precursor. Previously it was assumed that plant steroids have cycloartenol as precursor?

Line 286 Amaryllidaceae plants mainly contain steroidal saponins at present, I do not understand the message of this sentence.

Line 294 Compared with Amaryllidaceae, Agave species are the main plants containing in 294 Agavaceae, and a few Agavaceae plants such as Camassia and Yucca species have also 295 identified steroidal saponins. I do not understand the message of this sentence.  

Line 447 to 456: Saponins are also found in dicots.

Line 470 plant name in italic

Author Response

Response to Reviewer 3 Comments

Point 1: As mentioned below linguistic challenges and the size of Figure 2 have prevented me from understanding this manuscript. I, therefore, suggest that the manuscript has to undergo severe linguistic revision before it can be evaluated.

Response 1:  Thank you for your comments. For linguistic revision, we are going to use the “Language Editing Services” of MDPI to revise and polish our manuscript. For the problem of Figure 2, we redraw Figure 2 and show the details as much as possible so that you and readers can better understand our manuscript. The newly modified figure 2 is as follows:

Figure 2

Point 2:  I miss a definition of the topics of this review steroids and saponins. In old textbooks of pharmacognosy, saponins were defined as compounds, which could hemolyze red blood cells and reduce the surface tension of aqueous solutions.

Response 2: Thank you for your comments and suggestions. We have added some definitions of steroids and saponins in the introduction, as followed:Saponins isolated from plants are considered to have a variety of properties, often acting to protect plants from pathogens and plant-feeding insects. For humans and animals, saponins can affect cell membrane permeability, dissolve red blood cells, have hypoglycemic activity, and can also affect cholesterol metabolism; they therefore have a wide range of applications in food, cosmetics, pharmaceuticals, and other fields [1,4].”.

Point 3:  Why are the nine representatives in Figure 1 selected?

Response 3: Thank you for your comments. According to the structural types of the core aglycones of steroidal saponins isolated from plants reported so far, we have finally determined eight representative types of steroidal saponins in Figure 1.

Figure 1

Point 4:  A general problem is illustrated in lines 60 to 73. The different needs of different plants. The word need indicates that the plants have chosen to make steroids. Probably evolution has given plants producing saponins some advantages enabling them to survive in contrast to other plants. Thus, saponin production probably is a stochastic developed property developed in successful species in contrast to failing species. Thus, the plants did not develop saponins to overcome stress but survived because they produced saponins.

Response 4:  Thank you for your comments. Your opinion is very original, which is also what we have been interested in and are exploring. We can only objectively explain that steroidal saponins have different functions in plants, not different needs. In order to avoid misunderstanding, we change this sentence to “the distribution and existing forms of steroidal saponins in plants are diverse, which may also be related to the different functions of steroidal saponins in different plants.”.

Point 5:  I assume that Figure 2 is the essential figure in this manuscript revealing intermediates and products for saponin biosynthesis.  Unfortunately, Figure 2 can not be read. It must be enlarged. As a consequence of this bad figure I can not evaluate paragraph 2.3.

Response 5:  Thank you for your suggestion. We are sorry for that, and we have carefully revised Figure 2 to show details as much as possible.

Point 6:  After huge enlarging, I realized that both the deoxyxylolose and the mevalonate pathway are mentioned as sources for squalene. Is that true? In general, triterpenoids are believed to drive from mevalonic acid.

Response 6:  Thank you for your comments. Yes. Both MEP and MVA pathways can produce IPP and be catalyzed to form squalene, which is a common biosynthesis precursor for triterpenoids and steroids. In plants, triterpenoids can be synthesized through MEP or MVA pathway.

Point 7:  Another problem for me is that cholesterol is mentioned as a precursor of saponins. Some textbook claim that triterpenes in plants are derived from cycloartenol. In paragraph 2.2 cycloartenol synthase is mentioned and it is stated (I guess) that cycloartenol is a precursor for cholesterol. Is this true?

Response 7: Thank you for your comments. Yes. Cycloartenol is a synthetic precursor of cholesterol and triterpenoids, which can be synthesized into triterpenoids and sterols through a series of enzymatic transformations. We have pointed out in paragraph 2.2 in the manuscript.

Point 8:  Why is the steroid skeleton depicted differently in the blue and pink parts of figure 2?

Response 8: Thank you for pointing out this. We have redrawn Figure 2. The biosynthetic pathway of cholesterol is depicted in pink, the biosynthetic pathway of steroidal saponins is marked in blue, they're different stages of steroidal saponins biosynthesis.

Point 9:  The review is stated to focus on monocots. Why is Fabaceae then mentioned?

Response 9: Thank you for your comments. Steroidal saponins mainly found in monocots, but also in a few eudicots, such as Fabaceae, fenugreek. Moreover, fenugreek is also a popular plant for studying steroidal saponins biosynthesis. Some key genes for steroidal saponin biosynthesis (CYP90B, CYP72A, etc.) have been identified by using fenugreek. Therefore, whether in the "biosynthesis" section or the "distribution of steroidal saponins in plants" section, we have mentioned Fabaceae, mainly fenugreek.

Point 10:  Table 1: An impressive table mentioning steroids in different plant species. In column 3 (Plant groups) Firstly I do not know what a plant group is, secondly why is the name of the genus mentioned when this also appears I column 2?

Response 10:  Thank you for your comments. The plant groups in our table refer to plant family, which we have corrected. The repeated genus names mentioned in the second column have been removed.

Linguistic problems:

Point 11:  Line 153 Diosgenin generated by cholesterol should be generated from.

Response 11: Thank you for your suggestion. We have changed the sentence ‘diosgenin generated by cholesterol’ to ‘diosgenin is generated from cholesterol’. (Line 219 of revised manuscript)

Point 12:  Line 123 Cholesterol/β-sitosterol are synthesized by cycloartenol, and might be synthesized from cycloartenol.

Response 12: Thank you for your suggestion. We have changed the sentence ‘Cholesterol/β-sitosterol are synthesized by cycloartenol’ to ‘Cholesterol/β-sitosterol are synthesized from cycloartenol’. (Line 185 of revised manuscript)

Point 13:  163 but this part of the research is controversial, but this suggested pathway is dubious.

Response 13: Thanks for your advice. We have changed this sentence to avoid misunderstanding, as follows: “β-sitosterol may also be catalyzed by a series of enzymes to form steroidal saponins, which needs further exploration.” (Line 231 of revised manuscript)

Point 14:  189 it is stated that plant steroids have cholesterol as a precursor. Previously it was assumed that plant steroids have cycloartenol as a precursor.

Response 14: Thank you for your comments. Steroids include sterols, steroidal saponins, steroidal alkaloids, etc. These substances are not collectively referred to as plant steroids, but described in terms of plant sterols and steroidal saponins respectively. What we described at the beginning is that cholesterol is synthesized from cycloartenol, and cholesterol is the precursor of steroidal saponins in plants. There is no contradiction between the two.

Point 15:  Line 286 Amaryllidaceae plants mainly contain steroidal saponins at present, I do not understand the message of this sentence.

Response 15: Thank you for pointing out this. I'm sorry for that, this sentence should be “At present, the main plant species containing steroidal saponins in Amaryllidaceae include Allioideae, Allium species.”. We have changed this sentence to avoid misunderstanding. (Line 384 of revised manuscript)

Point 16:  Line 294 Compared with Amaryllidaceae, Agave species are the main plants contain in 294 Agavaceae, and a few Agavaceae plants such as Camassia and Yucca species have also 295 identified steroidal saponins. I do not understand the message of this sentence. 

Response 16: Thank you for your comments. I'm sorry for that, this sentence should be “Compared with Amaryllidaceae, Agave species are the main plants containing steroidal saponins in Agavaceae, and a small amount of steroidal saponins have also been identified from a few Agavaceae plants such as Camascia and Yucca species”. We have changed this sentence to avoid misunderstanding. (Line 395 of revised manuscript)

Point 17:  Line 447 to 456: Saponins are also found in dicots.

Response 17: Thanks for pointing this out, we have added some representative eudicots to this sentence.

Point 18:  Line 470 plant name in italic

Response 18: Thank you for your advice. We have corrected it.

Round 2

Reviewer 1 Report

The manuscript improved greatly. However, I am still thinking that the figure must be bigger or present on a landscape page to be clear.

Author Response

Response to Reviewer 1 Comments

Point 1: The manuscript improved greatly. However, I am still thinking that the figure must be bigger or present on a landscape page to be clear.

Response 1: Thank you for your comments and suggestions. We have enlarged the Figure 2, and used a landscape page to show more details.

Reviewer 2 Report

I have checked the revised version. New text portions, directly relevant to the MS ideas, have appeared in all the sections of MS. I believe the manuscript has been significantly improved to warrant publication in International Journal of Molecular Sciences.

Author Response

Response to Reviewer 2 Comments

Point 1:  I have checked the revised version. New text portions, directly relevant to the MS ideas, have appeared in all the sections of MS. I believe the manuscript has been significantly improved to warrant publication in International Journal of Molecular Sciences.

Response 1: Thank you for your comments and suggestions. Thanks to this, our manuscript has been significantly improved.

Reviewer 3 Report

I admit that the language has been improved but it is obvious that a person who does not understand chemistry has performed the improvement. The language still is not publishable (see below).

The suggested definition is not satisfactory. The main property is not mentioned: water surface tension is missing. Dewick defines saponins as glycosides that even at low concentrations produce a frothing in an aqueous solution because they have surfactant and soap-like properties. [1]. This at least is a definition mentioning the main property of saponins the decrease of surface tension of water. This manuscript deals with steroidal saponins. The work is derived from the Latin word for soap sapo. The amphibolic properties probably is the main cause of many of the properties of saponins: they destroy the cell membrane by dissolving lipids in the membrane.

The authors still mention that cholesterol is the main precursor of steroidal saponins. In Dewick is a paragraph on Pytosterols discussing the differences between sterols in organisms making photosynthesis and other organisms [1]. This might be important.   

Figure 2 has been improved but is still difficult to read.

31 widely distributed in plants. This statement is contradicted by the manuscript which state that saponins mainly are found In monocot plants.

33 so they are also 33 called triterpenoid saponins and steroidal saponins. They are triterpenoid saponins and steroidal saponins.

Line 42 dissolve red blood cells , no they cause lysing of cells.

Line 80 saponins…. is should be saponins… are

Line 82: Pennogenin saponins: also rare in plants (Figure 1H) A verb is missing. Also rare in plants are they common elsewhere?

Line 89 In addition, it is also possible to introduce 89 double bonds between C-25 and C-27 of spirostanols, why should it not be possible?

Line 91 . According to the num- 91 ber of sugar units connected to the aglycone of steroidal saponins, they can be divided into monoglycoside, dioglycoside, trioglycoside and tetraoglycoside should be they are monoglycosides, diglycosides, triglycosides or tetraoglycosides

Line 128: as when being eaten by herbivorous animals, It is difficult to imagine that a plant eaten by a herbivore can produce saponins.

1.            Dewick, P. M., Medicinal Natural Products. 3rd ed.; John Wiley and Sons Ltd.: Chicester, UK, 2009.

Author Response

Response to Reviewer 3 Comments

Point 1: I admit that the language has been improved but it is obvious that a person who does not understand chemistry has performed the improvement. The language still is not publishable (see below).

Response 1: Thank you for your suggestions and comments. We have carefully revised the inaccurate description once again, hoping to meet the publishing standards.

Point 2: The suggested definition is not satisfactory. The main property is not mentioned: water surface tension is missing. Dewick defines saponins as glycosides that even at low concentrations produce a frothing in an aqueous solution because they have surfactant and soap-like properties. [1]. This at least is a definition mentioning the main property of saponins the decrease of surface tension of water. This manuscript deals with steroidal saponins. The work is derived from the Latin word for soap sapo. The amphibolic properties probably is the main cause of many of the properties of saponins: they destroy the cell membrane by dissolving lipids in the membrane.

Response 2: Thank you for your comments and suggestions. We have added main property (water surface tension) to the definition of saponins. (Lines 33-35)

Point 3: The authors still mention that cholesterol is the main precursor of steroidal saponins. In Dewick is a paragraph on Pytosterols discussing the differences between sterols in organisms making photosynthesis and other organisms [1]. This might be important. 

Response 3: Thank you for your comments. This statement is not specifically put forward by us. According to relevant research on steroidal saponins, cholesterol is the main precursor for the synthesis of steroidal saponins [1-3]. Because some researchers believe that β-sitosterol may also be the biosynthetic precursor of steroidal saponins [2], but there are few relevant reports. Therefore, we believe that cholesterol is the main precursor of steroidal saponins. Sonawane et al. mentioned in their article that cholesterol exists not only in animals but also in plants, moreover, cholesterol is also the key precursor of thousands of bioactive metabolites in plants [4], including steroidal saponins and steroidal glycolkaloids.

Point 4: Figure 2 has been improved but is still difficult to read.

Response 4: Thank you for your comments. For better reading, we have enlarged the figure 2 to show more details.

Point 5: 31 widely distributed in plants. This statement is contradicted by the manuscript which state that saponins mainly are found In monocot plants.

Response 5: Thank you for your comments. What we mentioned at the beginning of the manuscript is “saponins”, which contains various types of phytochemicals such as triterpenoid saponins and steroidal saponins. Saponins are a general term of a large class of plant natural products, which exist widely in plants indeed [1]. In addition, it is not contradictory to describe that steroidal saponin mainly exists in monocotyledon plants, and this point is mentioned by Moses et al in their review [1]. We started with a brief introduction of what kind of phytochemicals are called "saponins" and their distribution and function in plants as a background. Then, we introduced the main role of this review: "steroidal saponins". Moreover, our manuscript begins with the introduction of steroidal saponins at line 41.

Point 6: 33 so they are also 33 called triterpenoid saponins and steroidal saponins. They are triterpenoid saponins and steroidal saponins.

Response 6: Thank you for your advice. We have revised this sentence. (Line 31)

Point 7:  Line 42 dissolve red blood cells , no they cause lysing of cells.

Response 7: Thank you for your advice. We have corrected this inaccurate statement. (Line 38)

Point 8: Line 80 saponins…. is should be saponins… are

Response 8: Thank you for your advice. We have changed "is" to "are". (Line 77)

Point 9: Line 82: Pennogenin saponins: also rare in plants (Figure 1H) A verb is missing. Also rare in plants are they common elsewhere?

Response 9: Thank you for your suggestions and comments. We have changed this sentence to "Pennogenin saponins are also are in plants" (Line 78). We mean that pennogenin saponins and pseudospirostanol saponins are both rare in plants, which does not mean that they are common in other places.

Point 10: Line 89 In addition, it is also possible to introduce 89 double bonds between C-25 and C-27 of spirostanols, why should it not be possible?

Response 10: Thank you for your comments. This may also be possible, but we haven't found any research suggesting that double bonds can be introduced between C-25 and C-27 of spirostanol saponins. Therefore, we did not mention this in our manuscript. After all, this is a review, and all the contents need to be summarized according to the reported research results.

Point 11: Line 91. According to the number of sugar units connected to the aglycone of steroidal saponins, they can be divided into monoglycoside, dioglycoside, trioglycoside and tetraoglycoside should be they are monoglycosides, diglycosides, triglycosides or tetraoglycosides.

Response 11: Thank you for your advice. We have changed this sentence to “According to the number of sugar units connected to the aglycone of steroidal saponins, they are monoglycosides, dioglycosides, trioglycosides or tetraoglycosides, and the common sugar units are pyranose, rhamnose and arabinopyranose”. (Lines 88-91)

Point 12: Line 128: as when being eaten by herbivorous animals, It is difficult to imagine that a plant eaten by a herbivore can produce saponins.

  1. Dewick, P. M., Medicinal Natural Products. 3rd ed.; John Wiley and Sons Ltd.: Chicester, UK, 2009.

Response 12: Thank you for your comments. Yes, the plants containing steroidal saponins can produce saponins in response to attacks by plant-feeding insects. The word "herbivorous animals" in our manuscript should be exactly "herbivorous insects". To avoid misunderstanding, we revised the word. (Line 111)

References:

  1. Moses, T.; Papadopoulou, K.K.; Osbourn, A. Metabolic and functional diversity of saponins, biosynthetic intermediates and semi-synthetic derivatives. Crit. Rev. Biochem. Mol. Biol. 2014, 49, 439–462.
  2. Chen, Y.; Wu, J.; Yu, D.; Du, X. Advances in steroidal saponins biosynthesis. Planta 2021, 254, 1–17.
  3. Upadhyay, S.; Jeena, G.S.; Shikha; Shukla, R.K. Recent advances in steroidal saponins biosynthesis and in vitro production. Planta 2018, 248, 519–544.
  4. Sonawane, P.D.; Pollier, J.; Panda, S.; Szymanski, J.; Massalha, H.; Yona, M.; Unger, T.; Malitsky, S.; Arendt, P.; Pauwels, L.; et al. Plant cholesterol biosynthetic pathway overlaps with phytosterol metabolism. Nat. Plants 2016, 3, 16205.

Round 3

Reviewer 3 Report

I find the manuscript publishable now. You might ask a person skilled in scientific English to read it. As you can see, I have found a few points. On the other hand, I have only read the changes. Please do not send the manuscript to me again.

Finally, Fig. 1 can be read! I wonder if it is necessary to start the figure with glyceraldehyde and mevalonic acid. You could start with dimethylallylPP and isopentenylPP.

Line 33 skeletons partially connected: skeleton conjugated to a sugar moiety,

Line 38 affect cell membrane permeability or lysing of the cells,

Author Response

Response to Reviewer 3 Comments

Point 1: I find the manuscript publishable now. You might ask a person skilled in scientific English to read it. As you can see, I have found a few points. On the other hand, I have only read the changes. Please do not send the manuscript to me again.

Response 1: Thank you for your comments and suggestions. Thanks to this, our manuscript has been significantly improved.

Point 2: Finally, Fig. 1 can be read! I wonder if it is necessary to start the figure with glyceraldehyde and mevalonic acid. You could start with dimethylallylPP and isopentenylPP.

Response 2: Thank you for your comment and suggestion. Figure 1 is summarized by referring to the reported studies. Current studies have shown that MVA and MEP pathways are involved in the biosynthesis of steroidal saponins [1,2], so we have listed both. In that case, it is better to briefly introduce both of these two pathways.

Point 3: Line 33 skeletons partially connected: skeleton conjugated to a sugar moiety,

Response 3: Thank you for your suggestion. We have revised this sentence to “They are a group of compounds with complex chemical structures composed of triterpenoid or steroidal aglycone molecular skeleton conjugated to a sugar moiety”. (Lines 29-31)

Point 4: Line 38 affect cell membrane permeability or lysing of the cells,

Response 4: Thank you for your suggestion. We have revised this sentence. (Line 38)

  1. Chen, Y.; Wu, J.; Yu, D.; Du, X. Advances in steroidal saponins biosynthesis. Planta 2021, 254, 1–17.
  2. Upadhyay, S.; Jeena, G.S.; Shikha; Shukla, R.K. Recent advances in steroidal saponins biosynthesis and in vitro production. Planta 2018, 248, 519–544.
